# Antibiotic prescribing practices in community and clinical settings during the COVID-19 pandemic in Nairobi, Kenya

June Gacheri[1,2☯], Katie A. Hamilton[3☯], Peterkin Munywoki[1], Sinaida Wakahiu[1], Karen Kiambi[1], Eric M. Fèvre[1,3], Margaret N. Oluka[2], Eric M. Guantai[2], Arshnee Moodley[1,4], Dishon M. Muloi[1,3]*

1 International Livestock Research Institute, Nairobi, Kenya, 2 Department of Pharmacology, Clinical Pharmacy and Pharmacy Practice, University of Nairobi, Nairobi, Kenya, 3 Institute of Infection, Veterinary and Ecological Sciences, University of Liverpool, Liverpool, United Kingdom, 4 Department of Veterinary and Animal Sciences, University of Copenhagen, Frederiksberg C, Denmark

☯ These authors contributed equally to this work.
* D.Muloi@cgiar.org

**Data Availability Statement:** The data collected from the community pharmacies survey have been made publicly available and can be accessed at

## Abstract

The COVID-19 pandemic has significantly impacted healthcare systems, including antibiotic use practices. We present data on patterns of antibiotic dispensing and use in community and hospital settings respectively in Nairobi, Kenya during the pandemic. We conducted interviews with 243 pharmacies in Nairobi using a standardised questionnaire from November to December 2021. The data collected included demographic characteristics, antibiotic customers, types of antibiotics sold, and antibiotic prescribing practices. Additionally, we retrospectively reviewed health records for 992 and 738 patients admitted in COVID-19 and general wards at two large inpatient hospitals between April 2020 and May 2021, and January 2019 to October 2021, respectively. Demographic, utilisation of laboratory services, treatment, clinical, and outcome data were collected using a modified Global WHO Point Prevalence Surveys (Global-PPS) tool. Almost all pharmacies (91.4%) served customers suspected of having COVID-19 with a mean weekly number of 15.6 customers. All pharmacies dispensed antibiotics, mainly azithromycin and beta lactams to suspected COVID-19 infected customers. 83.4% of hospitalised COVID-19 patients received at least one antibiotic at some point during their hospitalisation, which was significantly higher than the 53.8% in general ward patients (p<0.001). Similarly, the average number of antibiotics administered to COVID-19 patients was higher than that of patients in the general ward (1.74 vs 0.9). Azithromycin and ceftriaxone were the most commonly used antibiotics in COVID-19 patients compared to ceftriaxone and metronidazole in the general wards. Only 2% of antibiotic prescriptions for COVID-19 patients were supported by microbiological investigations, which was consistent with the proportion of 6.8% among the general ward population. Antibiotics were commonly prescribed to customers and patients suspected of having COVID-19 either in community pharmacies or in hospital, without a prescription or laboratory diagnosis. These findings emphasize the crucial role of antibiotic stewardship, particularly in community pharmacies, in the context of COVID-19.

https://doi.org/10.17638/datacat.liverpool.ac.uk/2614. The pseudonymized patient data supporting this study's hospital findings is secured at the International Livestock Research Institute (ILRI). These data contain sensitive and potentially identifiable personal health information. As participants did not provide informed consent for public data sharing, and our Institutional Research Ethics Committee (IREC) approval prohibits public data deposition, external access to the database requires a formal, approved request from interested parties. Motivated and written requests for data access can be submitted to ILRI IREC (ILRIResearchcompliance@cgiar.org) for review and consideration.

**Funding:** This study was supported by the ILRI/BMZ One Health Research, Education, Outreach and Awareness Centre (OHRECA to DM) program and the British Society for Antimicrobial Chemotherapy (grant code: BSAC-COVID-67 to DM). The funders had no role in study design, data collection and analysis, decision to publish, or preparation of the manuscript.

**Competing interests:** The authors have declared that no competing interests exist.

## Introduction

The use of antibiotics is the main driver in creating selective pressure for the emergence of antimicrobial resistance (AMR) globally [1, 2]. Global antibiotic consumption increased by 40% between 2000 and 2018, especially in high income countries [3, 4]. Per capita consumption in low- and middle-income countries remains low but is growing rapidly, fuelled by economic growth, ease of access of antibiotic and high infection burden [4]. Antibiotics continue to be prescribed or sold for upper respiratory infections, which are most likely of viral aetiology [5, 6]. For example, almost half of outpatient antibiotics sold in the United States are prescribed inappropriately [7].

Much remains unknown about how the COVID-19 pandemic is directly (or indirectly) impacting overall levels of AMR, [8, 9] but it is becoming apparent that there is a low incidence of secondary bacterial infections associated with COVID-19 infections [9–11]. Furthermore, preliminary data suggest an increase in AMR during the pandemic, particularly for Gram-negative pathogens such as extended-spectrum β-lactamase-producing *Enterobacterales* and resistant Acinetobacter spp. A recent study reported a 15% increase in the rate of Gram negative resistant pathogens in the United States [12] and another systematic review reported that 60.5% of secondary infections among COVID-19 patients were resistant [8] Despite the low prevalence of bacterial co-infection, few studies from high income settings suggest that antibiotics were often empirically prescribed amongst COVID-19 patients both at hospital admission and during hospital stays, increasing the risk of AMR [9, 13]. There is little evidence, however, of how antibiotic prescription patterns in community and in hospitals changed during the pandemic in Low- and Middle-Income Countries [14] Urban populations in Nairobi, as in other similar settings, often cannot afford to pay for health services at public or private clinics [15]. Instead, they rely on community pharmacies as the primary level of outpatient care for consultation, diagnosis, and prescription of drugs including antibiotics. Previous studies have shown that more than half of all antibiotics available in community pharmacies in the city are purchased without a prescription, and inappropriate prescribing practices are common [16]. We hypothesize that the COVID-19 pandemic is generating additional pressures on the levels of antibiotic use in the community and healthcare settings in Nairobi where inappropriate use is already prevalent. The aim of this study was to describe the antibiotic prescribing patterns in community pharmacies and two hospitals in Nairobi, Kenya during the pandemic as a basis for supporting antibiotic stewardship interventions during the pandemic.

## Material and methods

### Ethics statement and consent to participate

The collection of data adhered to the legal requirements of the Government of Kenya. Ethical approval for human data collection was obtained from the ILRI Institutional Research Ethics Committee (ILRI-IREC2021/41). Written informed consent was obtained from community pharmacy participants and administrative approvals obtained from the study hospitals. Additional information regarding the ethical, cultural, and scientific considerations specific to inclusivity in global research is included in S1 File.

### Study design and setting

This study consisted of two components: a cross-sectional survey of community pharmacies that was conducted between November and December 2021, and a retrospective analysis of health records for patients admitted COVID-19 and general wards at two large hospitals in the Nairobi metropolitan area. Patient records were collected between April 2020 and May 2021

for COVID-19 wards and between January 2019 and October 2021 for general population wards.

## Selection and data collection in community pharmacies

A total of 243 community pharmacies were selected across 30 sublocations in Nairobi city using an approach as described by Muloi *et al* [17] to capture different socioeconomic diversity and maximize spatial distribution. Due to the lack of a registry of pharmacies, it was not possible to randomly sample pharmacies within each administrative area, therefore convenience sampling was used instead. However, we made efforts to ensure consistency in the selection of study pharmacies by randomly dropping Global Positioning System (GPS) points within each sublocation and recruiting the nearest pharmacies as study participants. The final distribution of selected pharmacies is shown in Fig 1, and the concentration of sampling points mirrors the density of the population across this urban city.

In the community pharmacy, study participants were pharmacists and pharmaceutical technologists working in the pharmacies, who held a bachelor's degree or a diploma in pharmacy, respectively. According to Kenyan regulations, both pharmacists and pharmaceutical technologists are authorized to sell antibiotics, but only on the prescription of an authorized prescriber. Only physicians, dentists and specialist pharmacists are authorized to prescribe antibiotics. In this study, we refer to all individuals who sell antibiotics in a pharmacy, regardless of their level of clinical training, as 'pharmacists' for ease of reference but also from a customer's perspective as the person providing advice in a pharmacy is also the individual making the treatment decision. We developed a draft survey questionnaire to collect data on demographic characteristics of the pharmacists, demographic and clinical characteristics of

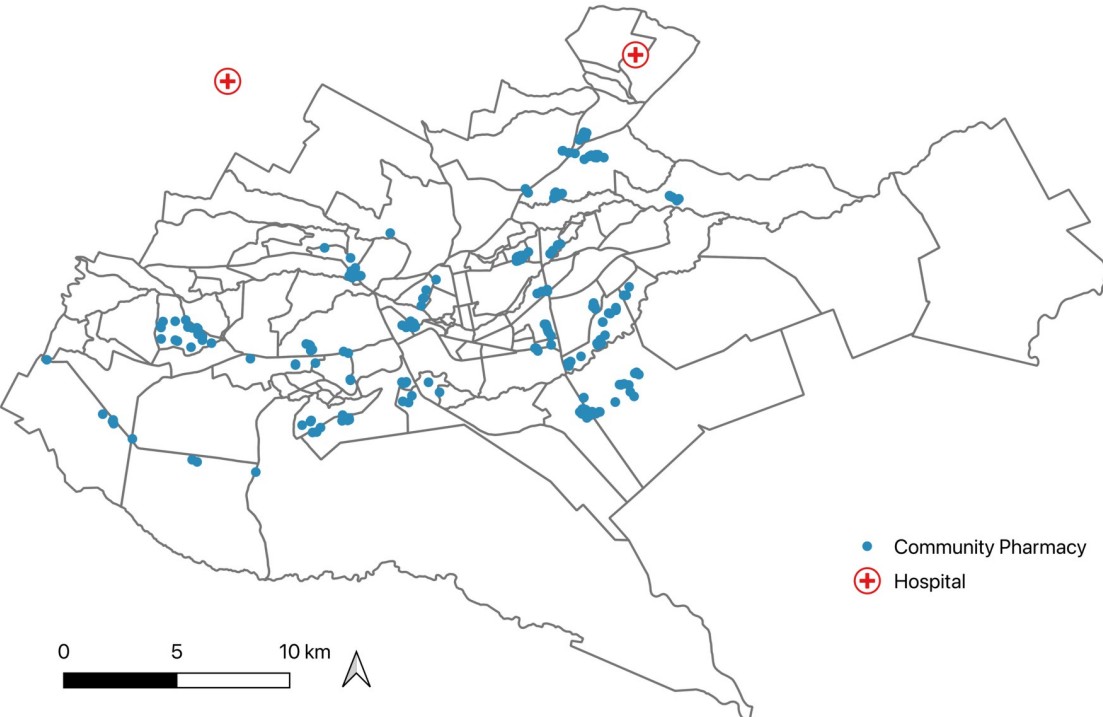

**Fig 1. Distribution of pharmacies and hospitals across the city of Nairobi.** Shape file was obtained from the Humanitarian Data Exchange (https://data.humdata.org/dataset/cod-ab-ken) and map created using QGIS version 3.28.

customers, types of antibiotics and other medicines sold and antibiotic prescribing practices. After pilot testing in five pharmacies, the questionnaire was refined, and finalized to meet the study's objectives. Four researchers administered the questionnaires, and the administration took 15 minutes. Twenty-one (8.6%) pharmacies that reported not having interacted with suspected COVID-19 individuals were excluded from the analysis.

## Selection and data collection in hospitals

This study included two hospitals: Tigoni Level 4 (with 100 beds) and Kenyatta University Teaching, Referral and Research Hospital (KUTRRH, with 650 beds). Both hospitals were designated COVID-19 treatment facilities, with KUTRRH handling patients with severe infections requiring high-dependency unit admission. We randomly reviewed and collected data from the clinical records of patients with a confirmed or presumptive COVID-19 diagnosis. Patients with a recorded negative SARS-CoV-2 test result were excluded from the analysis. Additionally, we collected data from a random subset of patients admitted to general population wards in both hospitals prior to and during the pandemic to assess antibiotic use patterns and practices in the absence of COVID-19. Patients in the general ward with respiratory infections were excluded from the study. In both COVID-19 and general wards, we collected demographic information of the patients, clinical information (including recorded COVID-19 severity), use of laboratory diagnostic testing, treatment decisions, and outcome data using a case report form adapted from the World Health Organization's (WHO) point-prevalence survey (PPS) forms [18]. Patient medical records were accessed between 29 November 2021 and 4 February 2022 in Tigoni Level 4 hospital and KUTRRH between 12 April 2022 and 29 April 2022.

## Statistical analysis

Categorical variables are presented as frequencies and percentages compared using the Chi-square-test, Fisher's exact test or Binomial proportion test. Continuous variables are presented as means, medians, standard deviations (SDs), quartiles, and ranges depending on data distribution. Hospital antibiotics were classified based on the Anatomical Therapeutical Chemical Classification (ATC Classification) and the WHO Access, Watch, and Reserve (AWaRe) classification. Diagnoses/indications for which antibiotics were prescribed in the general ward were recorded following prespecified PPS indication codes. Co-occurrence patterns among pairs of prescribed antibiotics were analysed based on the Jaccard Distance Matrix of the antibiotic data and represented visually as heatmaps.

To investigate the potential predictors of antibiotic use amongst inpatients both in COVID-19 and separately in the general ward, we used a Poisson general linear model (GLM) with counts of the individual antibiotics used in each patient as the outcome variable. Predictors analysed included age, sex, number of comorbidities, and length of hospitalisation. The hospital from which data were collected was included as a confounding factor in the models. Analyses were performed using the lme4 package [19] and significance was determined using Wald $\chi^2$-tests via car package [20]. All analysis was conducted in R v4.2.0, and P values of <0.05 were considered significant.

## Results

### Demographic and clinical characteristics

A total of 243 pharmacists were surveyed across the city of Nairobi. The median pharmacist age was 30 years (range, 27–66) and 117 (48.1%) of 243 participants were female. Of the

**Table 1. Characteristics of respondent community pharmacists.**

| Item | Total (%) |
|---|---|
| Number of pharmacies | 243 |
| Median age (range) | 30 (27–66) years |
| Gender | |
| Female | 117(48.1%) |
| Male | 126(51.9%) |
| Highest educational level | |
| Secondary | 3(1.2%) |
| Certificate | 10(4.1%) |
| Diploma | 163(67.1%) |
| Degree | 66(27.2%) |
| No formal education | 1(0.4%) |
| Clinical Training | |
| Yes | 239(98.4%) |
| No | 4(1.6%) |
| Role in the pharmacy | |
| Owner | 71(29.2%) |
| Worker | 172(70.8%) |

interviewed pharmacists, more than two-thirds were employees (n = 172, 70.8%), while the remainder owned the pharmacies. Almost all pharmacists reported having some form of clinical training (Table 1).

We analysed data from 992 patients admitted to COVID-19 wards and 738 in the general ward respectively from two hospitals. The median age of patients in the COVID-19 wards and the general ward was 54 and 35 years, respectively. We observed that 52.1% of COVID-19 patients exhibited mild symptoms, and COVID-19 diagnosis was frequently done using a rapid antigen test (53.6%). Comorbidities were present in 444 (44.8%) of patients in the COVID-19 wards and 215 (29.1%) of patients in the general ward (Table 2).

## Patterns of community antibiotic prescribing

Overall, 91.4% (222/243) of pharmacies reported serving a customer with a suspected COVID-19 infection, with a mean weekly number of 15.6 customers (range, 1–500). Most of the customers (86.5%, n = 192) in the 222 pharmacies did not have a confirmed COVID-19 diagnosis but showed symptoms commonly associated with the disease, such as sore throat (30.2%, n = 67), runny nose (25.7%, n = 57), fever (19.4%, n = 43) and cough (11.3%, n = 25). All pharmacies reported selling one or more antibiotics to customers suspected of having COVID-19, with 81.5% (n = 181) of pharmacies reporting that they prescribed an antibiotic without asking for a prescription. Azithromycin, amoxicillin-clavulanic acid, cefuroxime and amoxicillin were reported as the four most commonly sold antibiotics to customers suspected of having COVID-19 by 99.1%, 77%, and 64.9% of the pharmacies, respectively (Fig 2A). Two of the antibiotics, namely azithromycin and cefuroxime, are in the 'Watch' category by World Health Organization.

Azithromycin and amoxicillin-clavulanic acid were frequently co-prescribed (Jaccard similarity coefficient 0.67) followed by levofloxacin and ciprofloxacin (Jaccard similarity coefficient 0.25) (Fig 3A). The number of antibiotics prescribed to a customer did not differ by the pharmacist's clinical training status, or if the presenting customer had a confirmatory COVID-19 diagnosis or not (p>0.05).

**Table 2. Characteristics of patients admitted in COVID-19 and general wards.**

| Item | Tigoni n(%) | KUTRRH n(%) | Total (%) |
|---|---|---|---|
| **COVID-19 ward** | | | |
| Number of patients | 547 (55.1%) | 445 (44.9%) | 992 |
| Gender | | | |
| Female | 248(45.3%) | 155(34.8%) | 403(40.6%) |
| Male | 299(54.7%) | 290(65.2%) | 589(59.4%) |
| Median age (range) | 54 (2 months—99 Years) | | |
| COVID-19 severity | | | |
| Severe | 336(61.4%) | 139(31.2%) | 475(47.9%) |
| Mild | 211(38.6%) | 306(68.8%) | 517 (52.1%) |
| Diagnostic Method | | | |
| PCR | 66(12.1%) | 394(88.5%) | 460(46.4%) |
| Antigen Test | 481(87.9%) | 51(11.5%) | 532(53.6%) |
| Comorbidities present | 255(46.7%) | 189(42.5%) | 444(44.8%) |
| **General ward** | | | |
| Number of patients | 271 (36.7%) | 467 (63.3%) | 738 |
| Gender | | | |
| Female | 154 (56.8%) | 222 (47.5%) | 376 (50.9%) |
| Male | 117 (43.2%) | 245 (52.5%) | 362 (49.1%) |
| Median Age (range) | 35 (1 day– 100 years) | | |
| Surgical history | | | |
| Had surgery | 32(11.8%) | 79(16.9%) | 111 (15%) |
| No history | 223(82.3%) | 290(62.1%) | 513 (69.5%) |
| Unknown | 16(5.9%) | 98(21%) | 114 (15.4%) |
| History of catheterisation | 27(21.6%) | 101(10%) | 128 (17.3%) |
| Comorbidities present | 38(14%) | 177(37.9%) | 215 (29.1%) |

## Patterns of antibiotic use in hospitals

83.4% (827/992) of the admitted COVID-19 patients received one or more antibiotics at some point during their hospitalisation. This frequency of antibiotic use was significantly higher in the COVID-19 ward compared with general ward patients where only 53.8% (397/738) of admitted patients received an antibiotic (p<0.001, 95%CI: 0.25–0.34, Binomial Proportion test). Similarly, COVID-19 patients received a higher average number of different types of antibiotics (mean: 1.74, range: 0–5) than patients in the general ward (mean: 0.9, range: 0–6).

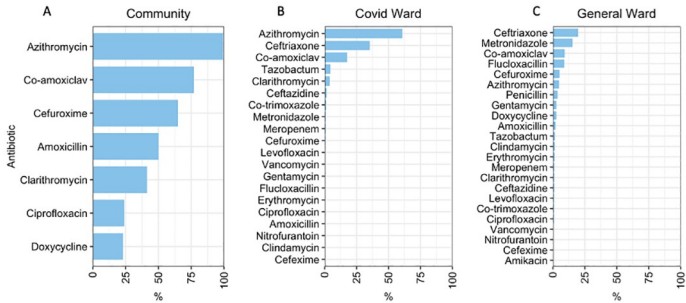

**Fig 2.** A-C: Antibiotics prescribed for A) customers in the community pharmacies, B) patients admitted in COVID-19 wards, and C) patients admitted in general wards.

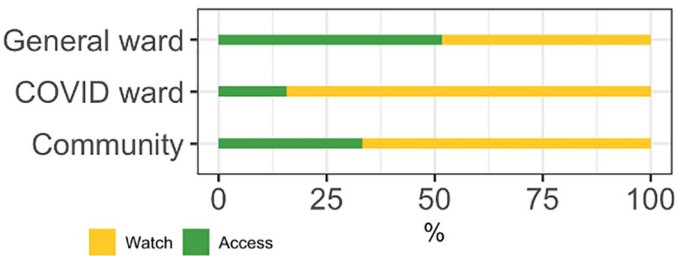

**Fig 3.** Co- prescription patterns of antibiotics in the A) community pharmacies, B) COVID-19 ward, C) general ward. The greater the intensity of colour shading, the greater the correlation of antimicrobial use measured by Jaccard distance.

Antibiotic use among both COVID-19 and general ward patients was significantly higher at Tigoni Hospital than in KUTRRH (COVID-19 ward; Tigoni 84.3%, KUTRRH 69.4%, P<001, OR = 2.4, 95% CI 1.7–3.2; general ward Tigoni 67.2%, KUTRRH 46%, P<0.001, OR = 2.4, 1.76–3.28, Fishers Exact Test).

Our model analysing factors associated with antibiotic use among COVID-19 patients demonstrated that older patients (OR: 1.2, 95% CI: 1.1–1.2, p<0.001) and those with multiple comorbidities (OR: 1.1, 95% CI: 1.0–1.1, p = 0.03) were significantly more likely to receive more antibiotics. Azithromycin (60.8%) was the most commonly prescribed antibiotic among COVID-19 patients, followed by ceftriaxone (35.4%) and amoxicillin-clavulanic acid (17.5%) (Fig 2B). The strongest correlation for co-prescription was between azithromycin and ceftriaxone (Jaccard similarity coefficient 0.38), followed by amoxicillin-clavulanic acid and azithromycin (Jaccard similarity coefficient 0.21) (Fig 3B). 84.3% (1082/1284) of antibiotic prescriptions were 'Watch' antibiotics, and 15.7% (202/1284) were from the 'Access' category (Fig 4). Stratification by hospital showed that Tigoni had a significantly higher percentage of antibiotic prescriptions from the WHO 'Watch' list compared to KUTTRH (OR: 1.31, Tigoni 55%, KUTTRH 45%, 95% CI 1.15–1.49, p<0.001, GLM).

The indication for prescribing antibiotics to COVID-19 patients based on the medical records obtained was for the management of secondary bacterial infections associated with COVID-19. However, microbiological investigations to identify the causative bacterial pathogens were conducted in only 2% (20/992) of admitted COVID-19 patients. Of all antibiotic prescriptions (n = 630) for patients in the general ward, ceftriaxone was the most frequently prescribed at 19.6% (n = 145), followed by metronidazole (15.4%, n = 114), amoxicillin-clavulanic acid (9.2%, n = 68), and flucloxacillin (8.8%, n = 65) (Fig 2C). Ceftriaxone and metronidazole were systematically co-prescribed (Jaccard similarity coefficient 0.22) in the general ward. 48.4% (282/583) of antibiotic prescriptions were from WHO AWaRe 'Watch' class, and

**Fig 4. Percentage of total antibiotic use by COVID-19 patients by WHO AWaRe classification by study site.**

**Table 3. Diagnoses for which antibiotics for treatment were prescribed in patients admitted in the general ward in the two study hospitals.**

| Diagnosis Code | Frequency | Percentage |
|---|---|---|
| SST | 65 | 16.4 |
| Proph BJ | 50 | 12.6 |
| UNK | 55 | 13.9 |
| CNS | 33 | 8.3 |
| Proph OBGY | 31 | 7.8 |
| IA | 28 | 7.1 |
| SEPSIS | 26 | 6.5 |
| Proph GI | 23 | 5.8 |
| NEO-MP | 15 | 3.8 |
| GI | 10 | 2.5 |
| BJ | 9 | 2.3 |
| Bron | 8 | 2 |
| Malaria | 8 | 2 |
| Cys | 6 | 1.5 |
| CVS | 5 | 1.3 |
| MP-GEN | 4 | 1 |
| Pneu | 4 | 1 |
| BAC | 3 | 0.8 |
| Proph CNS | 3 | 0.8 |
| ENT | 2 | 0.5 |
| Proph ENT | 2 | 0.5 |
| Pye | 2 | 0.5 |
| HIV | 1 | 0.3 |
| Proph CVS | 1 | 0.3 |
| Proph RESP | 1 | 0.3 |
| PUO | 1 | 0.3 |
| URTI | 1 | 0.3 |

51.6% (301/583) were from the 'Access' class (Fig 3C). The proportions of each WHO AWaRe category used did not differ significantly between hospitals (p>0.05, GLM). Microbiological investigation for causative bacteria was only conducted in 6.8% (27/397) of admitted patients. The main indications for prescribing antibiotics were skin and soft tissue infections (16.4%, 65/397), followed by prophylaxis for obstetric or gynaecological surgery (12.6%, 50/397), and infections of the central nervous system (8.3%, 33/397) (Table 3). A total of 13.9% (55/397) of patients who received antibiotics did not have a recorded diagnosis requiring antibiotic therapy.

SST, Cellulitis, wound including surgical site infection, deep soft tissue not involving bone; PROPH BJ, Prophylaxis for Skin and Soft Tissue, for plastic or orthopedic surgery; UNK Completely Unknown Indication; CNS, Infections of the Central Nervous System; PROPH OBGY, Prophylaxis for obstetric or gynecological surgery; IA, Intra-Abdominal sepsis including hepatobiliary, intra-abdominal abscess; SEPSIS, Sepsis of any origin, sepsis syndrome or septic shock with no clear anatomic site; PROPH GI, Gastro-Intestinal tract surgery, liver or biliary tree, gastro-intestinal prophylaxis in neutropenic patients or hepatic failure; NEO-MP, Medical prophylaxis for Newborn risk factors; GI, Gastro-Intestinal infections; BJ, Bone or Joint Infections such as Septic arthritis and osteomyelitis; BRON, Acute Bronchitis or

exacerbations of chronic bronchitis; CYS, Lower Urinary Tract Infection (cystitis); CVS, Cardiovascular system infections such as endocarditis, endovascular device; MP-GEN, antibiotic is used as medical prophylaxis in general, without targeting a specific site, PNEU, Pneumonia; BAC, Bacteremia or fungaemia with no clear anatomic site and no shock; PROPH CNS, Prophylaxis for CNS; ENT, Therapy for Ear, Nose, Throat infections including mouth, sinuses, larynx; Proph ENT Prophylaxis for Ear, Nose, Throat; PYE, Upper Urinary Tract Infection including catheter related UTI and pyelonephritis; HIV, Human immunodeficiency virus; Proph CVS, Cardiac or vascular surgery, endocarditis prophylaxis; Proph RESP, Pulmonary surgery and prophylaxis for respiratory pathogens; PUO, Pyrexia of Unknown Origin; URTI Upper Respiratory Tract viral Infections including influenza.

## Discussion

This study describes antibiotic prescribing patterns and practices in community and clinical settings in the city of Nairobi, Kenya before and during the COVID-19 pandemic. Antibiotics were commonly prescribed to COVID-19 patients either in community pharmacies or in hospital, without a prescription (in the community) or laboratory diagnosis. The proportion of antibiotics prescribed was higher in COVID-19 wards than in the general wards. These data support the continued need for developing antibiotic stewardship programmes targeting both the community pharmacies and hospitals.

Although there was no supporting evidence of bacterial infections in customers suspected of having COVID-19 presenting at a community pharmacy, all pharmacies sold antibiotics to them. Additionally, 83.4% of COVID-19 patients who were admitted to hospitals received at least one antibiotic kind, significantly higher than those in the general wards (53.8%). This high rate of antibiotic use is unsurprising given the inappropriate and empiric prescribing of antibiotics without proper diagnosis or prescription, which is prevalent in both community and hospital settings in Kenya even outside of the pandemic period [17, 21]. This finding is consistent with previous studies describing high rates of inappropriate antibiotic prescription in community pharmacies in Indonesia [22], Tanzania and Uganda [23], Saudi Arabia [24] and Russia [25], as well as among the general public to prevent or manage COVID-19 [14, 26, 27]. A recent metanalysis of global studies reported a pooled prevalence of 74.6% antibiotic usage amongst hospitalised COVID-19 patients [28]. These findings of frequent empiric antibiotic use can be partly attributed to clinical uncertainty, the acute nature and severity of the infection and the Kenyan national treatment guidelines that, at the time, recommended the use of antibiotics to manage COVID-19 cases [29, 30]. Similar observations of increased frequency of antibiotic use are evident globally during winter months [31] and highlight challenges of managing acute epidemics like COVID-19 and underscores the need for evidence-based decision-making to optimize treatment outcomes.

Given the rising occurrence resistance to 'Watch' antibiotics—generally broader spectrum antibiotic classes with critical clinical relevance—we examined their usage within our dataset. The proportion of prescription of 'Watch' antibiotics was much higher in the community and in hospitalised COVID-19 patients than the 'Access' antibiotics. In general, azithromycin, amoxicillin with or without clavulanic acid, and ceftriaxone were consistently and positively co-prescribed in both community pharmacies and hospitals. Although our study did not explicitly compare antibiotic usage before and during the pandemic, we demonstrate an apparent increase in the prescription of azithromycin. Globally, there was an increase in azithromycin prescription, singly or in combination with hydroxychloroquine, due to the perceived antiviral or anti-inflammatory effects against SARS-CoV-2 infection [32] which was later disputed [33, 34]. Although our study was not designed to investigate antibiotic resistance during

the pandemic, we hypothesize that the increase in empiric azithromycin use could serve as a strong predictor for increases in both macrolide resistance and resistance to other antibiotic classes, due to cross-resistance or co-selection [35].

Our analysis revealed a higher prevalence of antibiotic use among hospitalized COVID-19 patients who were older or those with multiple comorbidities. This finding aligns with existing literature that older patients and those with high prevalence of comorbidities are more likely to experience severe infections, predisposing them to antibiotic treatment. Therefore, emphasizing the need for additional investigations to improve our understanding of the antibiotic use in high-risk population groups. Our study also revealed that microbiological investigations were present in only 2% of antibiotic prescriptions for COVID-19 patients in the hospital, and similarly in the general ward (6.8%). This finding aligns with a previous study of fourteen public hospitals where only 1% of antibiotic prescriptions had documented microbiological investigations [21]. The low utilization of microbiological testing in patients with COVID-19 has been reported elsewhere, likely due to the immense clinical and laboratory pressure (increased COVID-19 testing) and staff shortages during the peak of the pandemic [11, 36]. To optimize antibiotic use, integrated approaches such as developing or strengthening antibiotic stewardship programs and guidelines, and updating the knowledge of healthcare workers within hospitals should be considered.

Our study has several limitations. In the community survey, we relied on reported sales patterns from pharmacy staff to analyse antibiotic use, instead of directly tracking individual prescriptions or reviewing medical/sales records as these were unavailable. This indirect approach limits our ability to definitively determine whether the prescriptions were appropriate or empiric, as we lack direct linkage to diagnoses or medical/sales records. Relatedly, our definition of a COVID-19 case was based on presumptive diagnoses by pharmacy staff based on clinical symptoms presented by patients, or the few patients who had a confirmed diagnosis. We acknowledge that these symptoms were not specific to COVID-19, which raises the possibility of staff overestimating the weekly number of COVID-19 patients. The hospitals included in our study were located in the city of Nairobi, where the prevalence of COVID-19 was highest in Kenya, limiting the generalizability of our findings to rural settings. Owing to the dynamic nature of the pandemic, characterised by frequent changes in clinical guidelines, it is possible that some of the findings in our study have changed. Lastly, data collection was conducted in the initial months of the pandemic or during peaks of infection waves. The timing of data collection varied across different components of the study, with the community study and COVID-19 ward thread being embedded in the pandemic period. On the other hand, the initial time point for the general population wards included a period prior to the onset of the pandemic. These different time points were selected based on convenience, as field research activities during the early stages of the pandemic were subject to restrictions.

## Conclusion

We characterized antibiotic prescribing in community pharmacies and hospitals during the COVID-19 pandemic to identify tractable targets for antibiotic stewardship. We show evidence of high rates of inappropriate antibiotic prescribing, without diagnostic support for bacterial infections, both in community and hospital settings, and disproportionately higher antibiotic use in COVID-19 patients that than those in general hospital populations. We observed, similar types of antibiotics prescribed in both community and clinical setting with minor variations in the general ward hospital populations. The long-term impact of COVID-19 on antibiotic use and AMR levels remains unknown; however, the possibility of it becoming endemic and virulent as the common cold, which is a common reason of antibiotic use,

underscores the need for sustained antibiotic stewardship programs. Stewardship initiatives aimed at optimizing antibiotic use should include community pharmacies, which are ubiquitous, as well as frontline healthcare providers, yet neglected by health authorities, in most LMICs.

## Supporting information

**S1 File. PLoS inclusivity in global research checklist.**
(DOCX)

## Author Contributions

**Conceptualization:** Katie A. Hamilton, Eric M. Fèvre, Dishon M. Muloi.

**Formal analysis:** June Gacheri, Katie A. Hamilton, Eric M. Guantai, Dishon M. Muloi.

**Funding acquisition:** Dishon M. Muloi.

**Investigation:** June Gacheri, Katie A. Hamilton, Peterkin Munywoki, Sinaida Wakahiu, Karen Kiambi, Dishon M. Muloi.

**Methodology:** June Gacheri, Katie A. Hamilton, Peterkin Munywoki, Sinaida Wakahiu, Karen Kiambi, Eric M. Fèvre, Margaret N. Oluka, Eric M. Guantai, Arshnee Moodley, Dishon M. Muloi.

**Supervision:** Margaret N. Oluka, Arshnee Moodley, Dishon M. Muloi.

**Validation:** Katie A. Hamilton.

**Writing – original draft:** June Gacheri, Dishon M. Muloi.

**Writing – review & editing:** June Gacheri, Katie A. Hamilton, Peterkin Munywoki, Sinaida Wakahiu, Karen Kiambi, Eric M. Fèvre, Margaret N. Oluka, Eric M. Guantai, Arshnee Moodley, Dishon M. Muloi.

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
