## [Decision Letter · Decision Letter 0]

6 Feb 2024

PGPH-D-23-02347

Antibiotic prescribing practices in community and clinical settings during the COVID-19 pandemic in Nairobi, Kenya

Dear Dr. Muloi,

Thank you for submitting your manuscript to PLOS Global Public Health. After careful consideration, we feel that it has merit but does not fully meet PLOS Global Public Health’s publication criteria as it currently stands. Therefore, we invite you to submit a revised version of the manuscript that addresses the points raised during the review process.

We look forward to receiving your revised manuscript.

Kind regards,

Raquel Muñiz-Salazar, Ph.D.

Academic Editor

Journal Requirements:

2. Please include a complete copy of PLOS’ questionnaire on inclusivity in global research in your revised manuscript. Our policy for research in this area aims to improve transparency in the reporting of research performed outside of researchers’ own country or community. The policy applies to researchers who have travelled to a different country to conduct research, research with Indigenous populations or their lands, and research on cultural artefacts. The questionnaire can also be requested at the journal’s discretion for any other submissions, even if these conditions are not met.  Please find more information on the policy and a link to download a blank copy of the questionnaire here: https://journals.plos.org/globalpublichealth/s/best-practices-in-research-reporting. Please upload a completed version of your questionnaire as Supporting Information when you resubmit your manuscript.

3. In the online submission form, you indicated that "The datasets generated and analysed in the current study are available from the corresponding author upon reasonable request."

3. Uploaded as supplementary information.

Additional Editor Comments (if provided):

Study limitations show weaknesses within a research design that may influence outcomes and conclusions of the research.

"It is important to address these questions in order to proceed with the revision process:

What is the operational definition of a suspected COVID-19 infection in a community context where there is no clinical data and prescription?

From what type of records did the researcher identify the symptoms commonly associated with the disease in the case of community pharmacies?

Reviewers' comments:

Reviewer's Responses to Questions

**Comments to the Author**

1. Does this manuscript meet PLOS Global Public Health’s publication criteria? Is the manuscript technically sound, and do the data support the conclusions? The manuscript must describe methodologically and ethically rigorous research with conclusions that are appropriately drawn based on the data presented.

Reviewer #1: Partly

Reviewer #2: Yes

2. Has the statistical analysis been performed appropriately and rigorously?

Reviewer #1: Yes

Reviewer #2: Yes

3. Have the authors made all data underlying the findings in their manuscript fully available (please refer to the Data Availability Statement at the start of the manuscript PDF file)?

Reviewer #1: Yes

Reviewer #2: Yes

4. Is the manuscript presented in an intelligible fashion and written in standard English?

Reviewer #1: Yes

Reviewer #2: Yes

5. Review Comments to the Author

Reviewer #1: Study limitations show weaknesses within a research design that may influence outcomes and conclusions of the research. The limitation mentioned (In the community survey, we extrapolated antibiotic use based on reported sales data, rather than directly tracking antibiotic prescriptions linked to specific diagnoses, making it challenging to determine whether the prescriptions were appropriate or empiric) in this research is very gross. Accordingly, it is difficult to generalize the study finding of community pharmacies where there is no recorded clinical data. It is very difficult to drive findings from sales data for this specific research.

It is better to answer the following questions

1. What is the operational definition of suspected COVID 19 infection in community context where there is no clinical data and prescription?

2. From what type of records the researcher got what symptoms are commonly associated with the disease in case of community pharmacies?

Reviewer #2: The article is well organized and very well written. It deals with the antibiotic misuse during the COVID-19 pandemic which impacts the AMR.

The only issue in the manuscript is the number of comorbidities in COVID-19 patients and in those in general wards. Now it is written that only 14% of COVID-19 patients had comorbidities which is not correct.

6. PLOS authors have the option to publish the peer review history of their article (what does this mean?). If published, this will include your full peer review and any attached files.

**Do you want your identity to be public for this peer review?** For information about this choice, including consent withdrawal, please see our Privacy Policy.

Reviewer #1: **Yes: **Guta Gurmessa Amenu (MD, MPH)

Reviewer #2: No

---

## [Decision Letter · Decision Letter 1]

10 Apr 2024

Antibiotic prescribing practices in community and clinical settings during the COVID-19 pandemic in Nairobi, Kenya

PGPH-D-23-02347R1

Dear Dr Muloi,

We are pleased to inform you that your manuscript 'Antibiotic prescribing practices in community and clinical settings during the COVID-19 pandemic in Nairobi, Kenya' has been provisionally accepted for publication in PLOS Global Public Health.

Best regards,

Raquel Muñiz-Salazar, Ph.D.

Academic Editor

Dear Authors,

I am pleased to inform you that, following a detailed and insightful review by two distinguished experts in the field, your manuscript "Antibiotic prescribing practices in community and clinical settings during the COVID-19 pandemic in Nairobi, Kenya" has been accepted for publication in Plos Global Public Health Journal. Your responsive and thorough revisions have significantly enhanced the manuscript's quality.

We believe your research will greatly interest our readership and will make a valuable addition to the scientific literature.

Reviewer Comments (if any, and for reference):

Reviewer's Responses to Questions

**Comments to the Author**

1. If the authors have adequately addressed your comments raised in a previous round of review and you feel that this manuscript is now acceptable for publication, you may indicate that here to bypass the “Comments to the Author” section, enter your conflict of interest statement in the “Confidential to Editor” section, and submit your "Accept" recommendation.

Reviewer #1: All comments have been addressed

Reviewer #2: All comments have been addressed

2. Does this manuscript meet PLOS Global Public Health’s publication criteria? Is the manuscript technically sound, and do the data support the conclusions? The manuscript must describe methodologically and ethically rigorous research with conclusions that are appropriately drawn based on the data presented.

Reviewer #1: Yes

Reviewer #2: Yes

3. Has the statistical analysis been performed appropriately and rigorously?

Reviewer #1: Yes

Reviewer #2: Yes

4. Have the authors made all data underlying the findings in their manuscript fully available (please refer to the Data Availability Statement at the start of the manuscript PDF file)?

Reviewer #1: Yes

Reviewer #2: Yes

5. Is the manuscript presented in an intelligible fashion and written in standard English?

Reviewer #1: Yes

Reviewer #2: Yes

6. Review Comments to the Author

Reviewer #1: This is a research with two study design addressing study challenges.

Reviewer #2: (No Response)

7. PLOS authors have the option to publish the peer review history of their article (what does this mean?). If published, this will include your full peer review and any attached files.

**Do you want your identity to be public for this peer review?** For information about this choice, including consent withdrawal, please see our Privacy Policy.

Reviewer #1: **Yes: **Guta Gurmessa Amenu

Reviewer #2: No
